# Identification and Allele Combination Analysis of Rice Grain Shape-Related Genes by Genome-Wide Association Study

**DOI:** 10.3390/ijms23031065

**Published:** 2022-01-19

**Authors:** Bingxin Meng, Tao Wang, Yi Luo, Ying Guo, Deze Xu, Chunhai Liu, Juan Zou, Lanzhi Li, Ying Diao, Zhiyong Gao, Zhongli Hu, Xingfei Zheng

**Affiliations:** 1State Key Laboratory of Hybrid Rice, Hubei Lotus Engineering Center, College of Life Sciences, Wuhan University, Wuhan 430072, China; 2016202040076@whu.edu.cn (B.M.); 2019202040069@whu.edu.cn (T.W.); 2019202040076@whu.edu.cn (Y.L.); ydiao@whu.edu.cn (Y.D.); huzhongli@whu.edu.cn (Z.H.); 2Institute of Food Crops, Hubei Academy of Agricultural Sciences, Wuhan 430064, China; guokkkyyy@163.com (Y.G.); dezexu@163.com (D.X.); zoujuan1010@163.com (J.Z.); 3Hunan Engineering Technology Research Center, Hunan Agricultural University, Changsha 410128, China; Chunhai.liu@stu.hunau.edu.cn (C.L.); lancy0829@163.com (L.L.)

**Keywords:** grain shape, GWAS, gene-based association analysis, haplotype analysis, allele combination analysis

## Abstract

Grain shape is an important agronomic character of rice, which affects the appearance, processing, and the edible quality. Screening and identifying more new genes associated with grain shape is beneficial to further understanding the genetic basis of grain shape and provides more gene resources for genetic breeding. This study has a natural population containing 623 indica rice cultivars. Genome-wide association studies/GWAS of several traits related to grain shape (grain length/GL, grain width/GW, grain length to width ratio/GLWR, grain circumferences/GC, and grain size/grain area/GS) were conducted by combining phenotypic data from four environments and the second-generation resequencing data, which have identified 39 important Quantitative trait locus/QTLs. We analyzed the 39 QTLs using three methods: gene-based association analysis, haplotype analysis, and functional annotation and identified three cloned genes (*GS3, GW5, OsDER1*) and seven new candidate genes in the candidate interval. At the same time, to effectively utilize the genes in the grain shape-related gene bank, we have also analyzed the allelic combinations of the three cloned genes. Finally, the extreme allele combination corresponding to each trait was found through statistical analysis. This study’s novel candidate genes and allele combinations will provide a valuable reference for future breeding work.

## 1. Introduction

Rice is an important food crop, and its production is of great significance to global food security, social stability, and economic development. 1000-grain weight is an essential factor affecting yield, and it can be inherited stably [1]. In addition, the factors determining 1000-grain weight can be divided into three aspects of grain shape: grain length, grain width, and grain thickness, and with the increase of grain length, grain width, and grain thickness, 1000-grain weight also increased [2]. In addition, grain shape has an essential effect on rice yield. It affects the appearance, processing, and edible quality of rice, which directly affects the market demand for rice. Across the globe, rice preferences vary markedly in grain shape: People in Europe, the United States, southern China, and Southeast Asia prefer slender rice, while people in Japan, the Korean Peninsula, and northern China prefer short, round rice [3]. In addition to grain shape, other appearance qualities of rice also include chalkiness, transparency, and color. Some studies showed that grain length was negatively correlated with chalkiness rate, while grain width was positively correlated [2]. For processing quality, in general, grain width and grain thickness were positively correlated with head rice rate, while grain length was negatively correlated. Grain shape has an essential effect on rice yield and quality traits. Therefore, identifying the genes that affect rice grain shape and elucidating their mechanism is an effective way to develop high-yield and high-quality rice.

Many factors affect the grain shape of rice, including the size and shape of the glume, the process of grain filling after flowering, and the development of grains. Genes involved in these pathways can directly or indirectly regulate the development of grain shape. The classification of cloned genes affecting grain shape generally shows the following ways to control rice grain shape: 1. endogenous hormone regulation, 2. G protein signaling pathway, 3. ubiquitin-proteasome pathway, 4. transcriptional regulation, 5. epigenetic pathway, 6. other regulatory pathways. These regulatory pathways are not independent of each other but often interweave together and act together. For example, *OsSCP46* interacts with ABA-induced protein DI19-1 and participates in the abscisic acid signaling pathway. *OsSCP46* coding a serine peptidase is a vital control factor for grain grouting and seed germination. Knocking out *OsSCP46* would reduce grain size, grain length, grain width, and 1000-grain weight [4]. *GW5* protein is a novel positive regulator of brassinolide signal transduction. It regulates the expression level of the brassinolide response gene and growth response. In addition, *GW5* encodes a calmodulin-binding protein and is a significant grain width gene. The grain size of *GW5* overexpressed mutants was slenderer than that of the wild type [5]. *GS3* encodes a γ subunit of G protein and is a significant grain length and weight gene. *GS3* plays a negative regulator role in regulating grain size. *GS3* does not directly restrict grain size but competes with *DEP1* and *GGC2* in binding G protein β subunits [6]. *OsDER1* is associated with the endoplasmic reticulum-associated protein degradation pathway. Overexpression or inhibition of *OsDER1* would activate the unfolded protein response, making rice more sensitive to ER stress and significantly increasing the ubiquitinated protein level. Inhibition of *OsDER1* expression would reduce both grain length and grain width [7]. Both *OsmiR396a* and *OsmiR396c* are micro RNAs that regulate the expression of *OsGRFs*, a growth regulator. Overexpression of *OsmiR396a* did not affect grain size. However, downregulation of *OsmiR396a* expression level would increase grain length and 1000-grain weight but decrease grain width [8]. *OsmiR396c* overexpressed mutants’ grain length, width, and weight would fall [9]. *JMJ703*, an active H3K4-specific demethylase in rice, can specifically reduce the methylation level of histone H3K4. The grain length, width, and thickness of the *JMJ703* mutant were reduced [10]. *OsCTPS1* encodes a CTP synthase. *OsCTPS1* interacts with tubulin, participates in microtubule function, promotes endosperm nuclear separation, influences endosperm early development, and positively regulates rice grain size and weight. Grain length, grain width, and grain thickness of *OsCTPS1* overexpressed lines were higher than those of wild type [11]. *OsARG* encodes an arginase, a key enzyme in arginine metabolism. *OsARG*-deficient mutants had smaller grains. *OsARG* plays a vital role in the panicle development of rice, especially under the condition of insufficient exogenous nitrogen. Therefore, *OsARG* can improve the nitrogen use efficiency of rice and is a potential target gene in crop improvement [12].

In this study, we have a natural population containing 623 indica rice cultivars with rich genetic diversity for GWAS (Genome-wide association studies) analysis. We combined rich phenotypic and genotypic data and adopted a strict MLM model for GWAS analysis. We further analyzed the results of GWAS using a variety of methods. Finally, we identified three cloned genes and seven new candidate genes. In addition, we have done allele combinations research among the three cloned genes through statistical analysis. The discovery of new genes will help us better understand grain shape regulation mechanisms. In addition, the new candidate genes and allele combinations found in this experiment will provide a valuable reference for future breeding work.

## 2. Results

### 2.1. Distribution and Correlation of Phenotype and Heritability of Grain Shape

In general, the phenotype of quantitative traits is normally distributed. This is because multiple genes with minor effects mainly control quantitative characteristics. However, in this study, the phenotypic distribution of grain length, grain width, grain length to width ratio, and grain circumference showed bimodal distribution except for the normal distribution of grain area (Figure 1). The distribution of phenotypes suggested that grain length, grain width, grain length to width ratio, and grain circumference were controlled by a major effect gene [13]. In contrast, grain area was controlled by multiple minor effect genes. Moreover, the wide range of phenotypic differences also indicates the rich genetic diversity of the study population. In terms of phenotypic correlation, grain length was significantly negatively correlated with grain width, while grain length was significantly positively correlated with other traits; Grain width was negatively correlated with grain length to width ratio and grain circumference, but positively correlated with grain area; Grain length-width ratio was positively correlated with grain area and grain circumference; Grain area was positively correlated with grain circumference (Figure 2). We also carried out Mahalanobis distance calculation and canonical variables analysis (CVA) for these grain shape traits, and the results were consistent with the above analysis (Table 1, Figure 3). In terms of heritability, the grain length to width ratio had the highest heritability (0.816), and the grain area had the lowest heritability (0.311). A high heritability indicates the stability of heredity, while a low heritability indicates that the character is greatly affected by the environment (Table 2).

### 2.2. Population Structure, Kinship, and LD Decay

As can be seen from the diagram of principal component analysis, scattered points are continuously distributed without evident clustering (Figure 4A). There were also no significant hot spots on the kinship map (Figure 4B). This indicates that our experimental population has no significant genetic structure and kinship. Combined with the distribution of phenotypes, it can be said that our experimental population fully conforms to the standard of the related population.

The LD decay distance determines the minimum number of molecular markers (Minimum molecular markers = genome size /LD decay distance) required for association analysis and the resolution of association analysis. As can be seen from the LD attenuation diagram, the LD attenuation distance is about 65 kb (Figure 4C). We have 2,416,716 SNPs, which is perfectly sufficient. In general, the richer the genetic diversity of the population, the shorter the LD attenuation distance, and vice versa [14]. Previously, it was reported that the LD decay distance of rice was about 130 kb [15]. Compared with this, our LD decay distance was smaller, and the population’s genetic diversity was richer. All of these will help us identify candidate genes.

### 2.3. Identification of Significant Loci for Related Traits through GWAS

For two years (2017, 2018) in Gongan/GA and Ezhou/EZ, Hubei Province, China, a total of 95 QTLs were detected. QTLs for grain width were the most (38), while QTLs for grain area were the least (2). 2017GA detected the most QTLs (51), while 2018EZ detected the least QTLs (39). Among the 95 QTLs, 40 QTLs were repeatedly detected, 15 QTLs regions with cloned genes, 24 QTLs with PVE ≥ 10% (Appendix A). Finally, 39 important QTLs were selected for further analysis (Table 3).

### 2.4. Candidate Genes Screen in Important QTL Regions

We have selected 39 important QTLs for further analysis involving four-grain shape traits: grain length, grain width, grain length to width ratio, and grain circumference. Through analysis, three cloned genes were identified: *GS3, GW5,* and *OsDER1*, seven novel candidate genes were identified: *Os02g0805100*, *Os02g0805400*, *Os02g0164600*, *CYP93G1*, *Os10g0344500*, *Os10g0344900,* and *Os02g0805100* (Figure 5). Most of these genes are pleiotropic and affect multiple traits.

In terms of grain length/GL, we have screened 4 QTLs (*qGL2*, *qGL3.2*, *qGL3.3*, *qGL5*) for analysis. Moreover, three (*qGL2*, *qGL3.3*, *qGL5*) of the four QTL intervals contained reported genes, 2 QTLs: *qGL3.3* and *qGL5* were detected repeatedly (Table 3). According to the functional annotation, there are 147 genes in these 4 QTLs intervals. In addition, according to the haplotype analysis, 15 genes belong to group I, 32 belong to group II. In the two groups, 15 genes were repeatedly detected, and, in total, 32 genes were detected, including two cloned genes: *GS3* and *GW5* (Figure 6A,B). *GS3* was detected in all environments, and *GW5* was detected except for 2018GA. Moreover, *GS3* also has a greater PVE value than *GW5* (Table 3), which coincides with previously reported that *GS3* has a major effect on grain length, while *GW5* has a minor impact on grain length. However, in the *qGL2* interval, we have not identified *OsmiR396a* and *OsmiR396c* (Table 3). In *qGL2*, four genes (*Os02g0804900*, *Os02g0805100*, *Os02g0805300,* and *Os02g0805400*) belong to group II. According to the functional annotation, *Os02g0804900* is identical to *RNRL2* [16]. *RNRL2* encodes a large ribonucleic acid reductase subunit involved in chlorophyll metabolism and regulates leaf color. *Os02g0805100* is an auxin/IAA gene (Figure 6C), *Os02g0805300* encodes an expressed protein, and *Os02g0805400* encodes a kelch repeat protein (Figure 6D). In the previous study, we know that auxin/IAA plays a vital role in regulating rice grain shape; Furthermore, one gene, *OsPPKL2*, encodes a protein phosphatase containing the kelch repeat domain and plays a positive regulator role in rice grain length regulation [17]. Therefore, we selected *Os02g0805100* and *Os02g0805400* as candidate genes based on their function correlation with cloned grain shape genes. Although *qGL3.2* had a large PVE value (20.37%, Table 3), only one gene, *Os03g0400600,* with the unknown function, was found in this interval.

In terms of grain width/GW, we have screened 10 QTLs (*qGW1.3*, *qGW2.1*, *qGW3.4*, *qGW4.1*, *qGW5.1*, *qGW5.2*, *qGW10.1*, *qGW10.7*, *qGW10.8*, *qGW10.12*) for analysis. Among the 10 QTLs, 6 QTLs (*qGW1.3*, *qGW3.4*, *qGW4.1*, *qGW5.1*, *qGW5.2*, *qGW10.1*) intervals contained cloned genes. Except for *qGW1.3* and *qGW5.2*, the other eight QTLs were repeatedly detected (Table 3). There were 343 genes in the 10 QTLs intervals, including four genes belonging to group I and 42 belonging to group II. A total of 42 genes were found between the two groups, including three cloned genes: *GS3*, *GW5,* and *OsDER1* (Figure 6E–G). *GS3* was detected in all environments, *GW5* was detected except for 2018GA, and *OsDER1* was detected only in 2018EZ. Furthermore, *GW5* had the highest mean PVE values (36.48%) (Table 3) among these genes, confirming that *GW5* was the dominant gene for grain width. However, for the remaining four cloned genes (*OsCTPS1*, *OsARG*, *JMJ703,* and *OsSCP46*) in the interval, through analysis, we have not identified them. In the gap *of qGW2.1*, we found five genes *Os02g0162000* (Cytochrome c oxidase), *Os02g0162500* (a ribosomal protein), *Os02g0162600* (Conserved hypothetical protein), *Os02g0164300* (galactosyltransferase), and *Os02g0164600* (Pentatricopeptide repeat domain-containing protein) belonging to group II. At the same time, we know that *OsSMK1* also encodes a pentatricopeptide repeat protein, *OS_smk1-1* homozygous mutant seeds were shriveled [18]. Therefore, we selected *Os02g0164600* (Figure 6H) as the candidate gene. During the interval *of qGW4.1*, we found 21 genes belonging to group II. *Os04g0101400* is identical *CYP93G1*, encoding cytochrome P450. But, in previous reports, it was not stated that *Os04g0101400* affects grain shape [19]. However, our analysis showed a significant correlation with grain width. In addition, we also found that there are three genes, *GL3.2*, *CYP78A13,* and *CYP724B1* [20,21] that encode cytochrome P450, all of which significantly affect grain shape. Finally, after haplotype analysis and functional annotation, we selected *Os04g0101400* (Figure 6I) as a candidate gene.

In terms of grain length to width ratio/GLWR, we have screened 22 QTLs for analysis. Among the 22 QTLs, 2 QTLs (*qGLWR3.3* and *qGLWR5.2*) contained cloned genes; Except for *qGLWR2.4*, *qGLWR2.6*, *qGLWR2.7*, *qGLWR10.2,* and *qGLWR10.4*, the other 17 QTLs were repeatedly detected (Table 3). There were 770 genes in the 22 QTLs intervals, including 15 genes belonging to group I and 51 genes belonging to group II. A total of 51 genes were found between the two groups, including two cloned genes: *GS3* and *GW5* (Figure 6J,K). Both *GS3* and *GW5* were detected in all environments, and both *GS3* and *GW5* had large PVE (39.32% and 41.66%, respectively) (Table 3) values and had important effects on grain length to width ratio. At the same time, we have not screened the cloned gene *OsDER1* within the interval of *qGLWR5.2*. In *qLWR1.2*, only one gene, *Os01g0589900* (Figure 6L), encodes a pentatricopeptide repeat protein. We selected *Os01g0589900* as the candidate gene. In the interval *of qGLWR10.10*, we found six genes *Os10g0343050* (expressed protein), *Os10g0343200* (membrane-associated DUF588 domain-containing protein), *Os10g0343400* (CSLF7 - cellulose synthase-like family F), *Os10g0343951* (Hypothetical conserved gene), *Os10g0344500* (MATE efflux family protein), and *Os10g0344900* (MATE efflux family protein) belonging to group II. At the same time, we know that *BIRG1* also encodes a MATE efflux family protein and functions as a chloride efflux transporter involved in mediating grain size [22]. Therefore, we selected *Os10g0344500* and *Os10g0344900* (Figure 6M,N) as candidate genes.

In terms of grain circumferences/GC, we have screened 3 QTLs: *qGC2*, *qGC3,* and *qGC5* for analysis. These three QTLs intervals all contain cloned genes associated with grain shape. While only one QTL *qGC3* was repeatedly detected (Table 3). There were 116 genes in the three QTLs intervals, including 15 genes belonging to group I and 27 genes belonging to group II. And a total of 27 genes were found between the two groups, including one cloned gene: *GS3* (Figure 6O). *GS3* was detected in all environments except for 2018GA. *MicroRNA OsmiR396a* and *OsmiR396c*, and the genes *GW5* and *OsDER1*, were not identified by screening. In *qGC2*, there is only one gene, *Os02g0805100* (Figure 6P); We already knew that *Os02g0805100*, which encodes auxin/IAA, was a candidate gene for grain length, and now we found that it pleiotropic, and it also a candidate gene for grain circumference.

### 2.5. Extreme Combination of Alleles for Each Trait

Due to the limitations of detected cloned genes, only allelic combinations of grain length/GL, grain width/GW, and grain length to width ratio/GLWR were analyzed; For GL and GLWR, we will explore the combination between the *GS3* and *GW5* alleles. And, for GW, we will examine the cross between the *GS3*, *GW5,* and *OsDER1* alleles.

Through allele analysis, we found that *GS3* had three alleles: Hap A (AATCT), Hap B (TGCTG), and Hap C (WRYYK) (Figure 7A). *GW5* has four alleles: Hap A (CG), Hap B (TA), Hap C (TG), and Hap D (YR) (Figure 7B). *OsDER1* also has three alleles: Hap A (T), Hap B (C), and Hap C (Y) (Figure 7C).

For grain length, Hap A (AATCT) has the most extended grain length in terms of *GS3*, Hap B (TGCTG), and Hap C (WRYYK) have nearly the same short-grain length. For *GW5*, Hap A (CG) has the most extended grain length, Hap B (TA) and Hap D (YR) have the shortest grain length. In our material, 6 of the 12 allelic combinations of *GS3* and *GW5* are included. The allele combination *GS3A-GW5A* of Hap A of *GS3* and Hap A of *GW5* has the most extended grain length. On the other hand, *GS3B-GW5B* has the shortest grain length (Figure 8A). Therefore, we hypothesized that the genetic effects of *GS3* and *GW5* on grain length were mainly additive.

For grain width, Hap B (TGCTG) has the maximum grain width in terms of *GS3*, Hap A (AATCT) has the minimum grain width. For *GW5*, Hap B (TA) has the maximum grain width, Hap A (CG) and Hap C (TG) have the minimum grain width. In *OsDER1*, Hap B (C) has the maximum grain width, Hap A (T) has the minimum grain width. Our material contains 9 of 36 allelic combinations of GS3, GW5, and OsDER1. The allele combination *GS3B-GW5B-GW5B* of Hap B of *GS3*, Hap B of *GW5,* and Hap B of *OsDER1* has the maximum grain width. On the other hand, *GS3A-GW5C-GW5A* has the minimum grain width (Figure 8B). The inheritance of grain width also showed the additive effect.

For grain length to width ratio, Hap A (AATCT) has the maximum GLWR in terms of *GS3*, Hap B (TGCTG) has the minimum GLWR. For *GW5*, Hap A (CG) has the maximum GLWR, Hap B (TA) has the minimum GLWR. Our material contains 6 of 12 allelic combinations of *GS3* and *GW5*. The allele combination *GS3A-GW5A* of Hap A of *GS3* and Hap A of *GW5* has the maximum GLWR. *GS3B-GW5B* has the minimum GLWR (Figure 8C). The inheritance of grain length to width ratio also showed the additive effect.

## 3. Discussion

Grain shape is an essential agronomic character of rice, which affects the yield and appearance, processing, and edible quality. What’s more, on a global scale, there is a marked difference in the grain shape of preferred rice. Therefore, grain shape also directly affects the market demand for rice. So far, many genes related to grain shape have been cloned, and their regulatory pathways have been classified, which provides an essential theoretical basis for breeding high-yield and good-quality rice. However, additional genes controlling grain shape remain to be identified, and the effective use of the cloned genes is still lacking.

The phenotypic distribution of GL, GW, GLWR, and GC showed bimodal distribution except for the normal distribution of GS (Figure 1). Therefore, we hypothesized that GL, GW, and GC have major effect genes [13]. As expected, we detected two cloned genes, *GS3* and *GW5*, which were the major genes affecting grain length and width, respectively [5,6]. As previously reported, *GS3*, *GW5,* and *OsDER1* regulate the grain shape through changes in expression level, and, in our haplotype analysis, they were mainly detected in group II. Therefore, it proves that our findings are consistent with previous reports [7,23,24]. Furthermore, we also found a significant negative correlation between GL and GW in the alleles of *GS3* and *GW5*. For example, *GS3* Hap A (AATCT) had the most extensive grain length but the smallest grain width, while *GW5* Hap B (TA) had the largest grain width but the most petite grain length (Figure 8). This finding fits well with the correlation between phenotypes (Figure 2).

After completing GWAS, we found that many QTL intervals contained cloned grain shape-related genes. Still, they were not detected through significance analysis, perhaps because the PVE of these QTL intervals was too small (Table 3). However, in some QTL intervals with large PVE, we still did not detect candidate genes. Even on the Manhattan plot, this region has a distinct peak, which we suspect is due to the type II error (false negative) caused by the MLM model. Compared with the GLM model, the accuracy of the MLM model is improved, but the detection efficiency is reduced. Using multiple detection models may effectively enhance detecting genes with minor effects. Due to environmental influences and gene-to-gene interactions, although many genes related to grain shape have been cloned, these genes still lack effective use. Next, to effectively utilize the genes in the grain shape-related gene bank, allelic combination analysis was performed. As it turns out, the inheritance of GL, GW, and GLWR mainly showed additive effects (Figure 8). Therefore, it is promising to design high-quality rice by pyramiding alleles of different genes. The seven new candidate genes were only obtained by genome-wide association study, gene-based association analysis, haplotype analysis, and functional annotation. However, their actual functions need to be further verified. Unfortunately, so far, validation work such as RT-PCR, transcriptome analysis, and transgenic experiments has not been completed.

## 4. Materials and Methods

### 4.1. Materials

The natural population used in this study consisted of 623 varieties collected worldwide. Among them, 323 accessions were selected from the 3K Rice Genome Project (3K R.G.P.) [25] (Appendix A).

### 4.2. Field Trials and Trait Measurements

From mid-May to the end of September in 2017 and 2018, we planted seeds in experimental fields in Gongan/GA Jingzhou and Ezhou/EZ, Hubei Province, China, respectively. Each variety was planted in 5 rows and 10 columns, and the distance between individual plants was 17 cm × 20 cm, with three replications. Field management methods are consistent with local management standards. When the rice was fully mature, seeds of 5 individual plants were collected. Wanshen SC-G seed copying apparatus was used to take photos and measure the related traits. The measured characteristics include grain length/GL, grain width/GW, grain length to width ratio/GLWR, grain circumferences/GC, and grain size/grain area/GS. For each trait, the mean value is used for GWAS analysis.

### 4.3. Statistical Analysis

We have used the software Graphpad Prism 9 to analyze the phenotypic data as follows: (1) Normal distribution test and (2) One-way analysis of variance. The analysis methods were the “Kolmogorov-Smirnov test” and “Brown-Forsythe and Welch ANOVA tests” respectively. At the same time, we also used canonical variables analysis and Mahalanobis distance analysis to analyze phenotypes.

### 4.4. Genome-Wide Association Study

#### 4.4.1. Genotyping

We extracted DNA from the leaves. Then, Covaris S2 (Covaris) breaks the DNA into ~500 bp fragments. NEBNext DNA Library Prep Reagent Set for Illumina (BioLabs) constructs the DNA library. Illumina Hiseq X10 platform was used to sequence the library. The reference genome was Os-Nipponbare-Reference-IRGSP-1.0 [26]. GATK was used to call single-nucleotide polymorphisms/SNPs [27]. SNPs with MAF ≥ 5% and missing rate ≤ 20% were retained. IMPUTE2 [28] is used for imputing missing genotypes, and 2,416,716 high-quality SNPs were finally obtained.

#### 4.4.2. Population Structure and Kinship Analysis

We used the R package “rMVP” [29] to calculate population structure (Q) and kinship (K). All SNPs are used in the calculation. The principal component analysis/PCA is used to evaluate population structure. The principal component analysis score and relationship matrix will be used in the mixed linear model (MLM) [30] below.

#### 4.4.3. Linkage Disequilibrium Analysis

We use the software “PopLDdecay” to calculate the linkage disequilibrium (LD) between pairs of markers [31]. Command: ^“^*r^2^*^”^, which squared the Pearson’s correlation coefficient (*r^2^*). When the correlation coefficient (*r^2^*) drops to half of its maximum value, the distance across the chromosome is called the LD decay distance [15].

#### 4.4.4. Genome-Wide Association Study and Candidate Genes Identification

GWAS is based on linkage disequilibrium (LD). It uses a large number of single nucleotide polymorphisms (SNPs) in the genome of the mapped population as molecular markers and combines with phenotypic variation of the population to analyze the correlation between target traits and molecular markers or candidate genes at the genome-wide level. GWAS analysis was performed using the R package “rMVP”. The operation steps of “rMVP” are as follows: (1) Data Preparation. Prepare phenotype and genotype data files and calculate population structure (Q) and kinship (K) based on the genotype files. (2) Data Input. Import the above four files into the operation. (3) Start GWAS. Important option parameters are model (“rMVP” offers three models: GLM, MLM, and FarmCPU.) and threshold (0.05 or 0.1, We chose 0.1). (4) Output. Here we have SNPs that are significantly correlated. We used a mixed linear model (MLM) to reduce false positives. MLM uses the Q and K matrix to adjust for cryptic relationships and other fixed effects.

Next, considering LD decay distance, we defined the interval of significantly associated SNP ± LD decay distance as a QTL. Here, referring to the previous report, we use the LD decay distance of 125 kb [15]. To reduce QTL redundancy, if there is overlap between QTL areas, they are combined into one QTL [32,33,34]. Next, we will select some important QTLs for further analysis, which must meet at least one of the following conditions: 1. They have been detected repeatedly in different environments. 2. Close to related genes that have been reported. 3. it contains successive distinct peaks. 4. Have large values of phenotypic variance explained /PVE (PVE ≥ 10%) [35]. For these important QTL regions, haplotype analysis and functional annotation of the genes within the regions were performed to screen candidate genes. The method of haplotype analysis was as follows: Firstly, the SNP was classified, the SNP that caused amino acid and splicing changes and had significant *p* value was classified into group I; SNPs with significant *p* values in the promoter region were assigned to the group II [36]. Then, these SNPs will be used for haplotype analysis (A gene haplotype with corresponding materials ≥ 6 will be retained), and genes with significant differences between haplotypes and functional correlation will be screened as candidate genes.

### 4.5. Allele Combination Analysis

#### 4.5.1. Allele Analysis of Different Genes

The definition of alleles is similar to that of haplotypes. Still, the difference is that the significant SNPs detected in different environments and different grain shape traits are different, so the haplotypes of the same gene may differ in different environments and other grain shape traits. However, alleles were different from haplotypes. If a gene affected multiple grain traits, SNPs of the gene that were significant in different environments and other characteristics were selected for allele analysis, and the materials were classified accordingly. Therefore, alleles of a gene are the same in different environments and for different characteristics [37]. Next, we matched the material with the mean phenotypic values of the relevant grain shape traits. Finally, through statistical analysis, find the extreme allele of each gene.

#### 4.5.2. Extreme Combination of Alleles for Each Trait

Each gene has an allele for an extreme phenotype for different grain traits; however, due to environmental influences and gene-to-gene interactions, the genotype of the optimal material is not necessarily a combination of the extreme allele of each gene. We use allelic combinations to classify materials, and if a combination has less than six materials, we discard the combination. Next, we matched the material with the phenotypic values of the relevant grain shape traits. And, through statistical analysis, find an extreme combination of alleles for each trait.

## 5. Conclusions

In 2017 and 2018, grain shape-related traits (grain length/GL, grain width/GW, grain length to width ratio/GLWR, grain circumferences/GC, and grain size/grain area/GS) of 623 indica rice cultivars were measured in Gongan/GA and Ezhou/EZ. And, in 2017, 623 rice were genotyped using second-generation resequencing technology. A genome-wide association study was performed based on the above genotypic and phenotypic data. A total of 39 important QTLs were screened out based on genome-wide association studies. In addition, haplotype difference analysis and functional annotation were performed on the genes in these candidate intervals. Finally, three cloned genes, *GS3*, *GW5,* and *OsDER1*, and seven new candidate genes *Os02g0805100*, *Os02g0805400*, *Os02g0164600*, *CYP93G1*, *Os10g0344500*, *Os10g0344900,* and *Os02g0805100* were found. We have also analyzed the allelic combinations of the three cloned genes *GS3*, *GW5,* and *OsDER1*. The results of this study will enrich the gene pool of grain shape, deepen the understanding of the regulation mechanism of grain shape, and provide a valuable reference for future molecular breeding work.

## Figures and Tables

**Figure 1 ijms-23-01065-f001:**
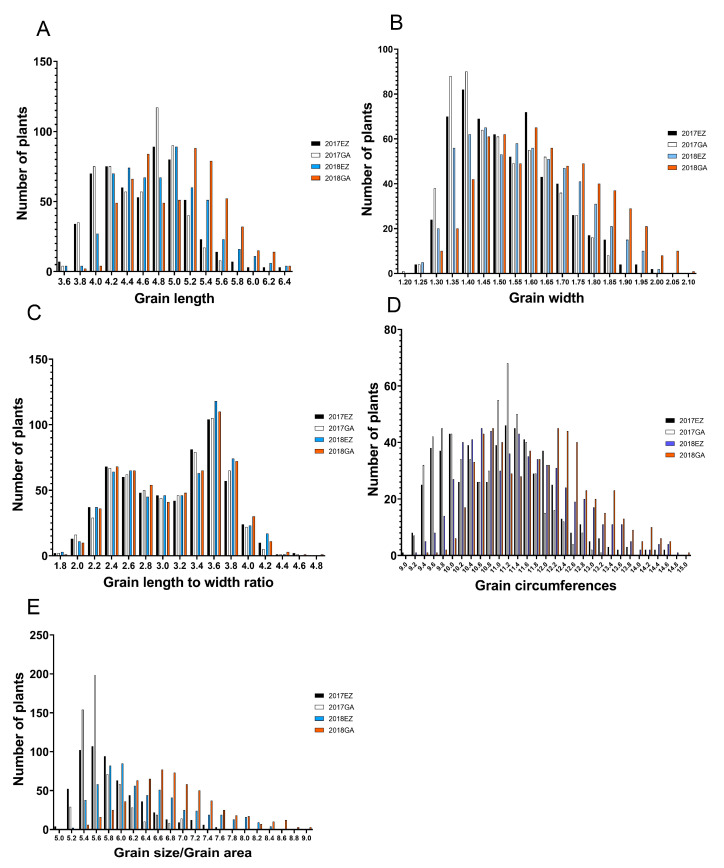
The phenotypic distribution. Phenotypic distribution of five-grain shape traits: (**A**) grain length/GL, (**B**) grain width/GW, (**C**) grain length to width ratio/GLWR, (**D**) grain circumferences/GC, and (**E**) grain size/grain area/GS) in 2017EZ, 2017GA, 2018EZ, and 2018GA. Gongan/GA, Ezhou/EZ.

**Figure 2 ijms-23-01065-f002:**
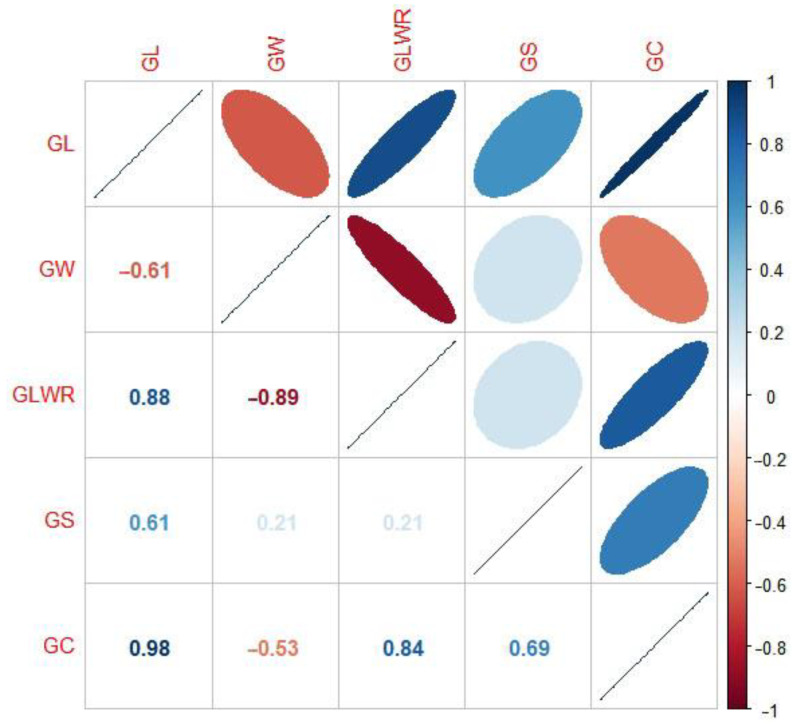
Phenotypic correlation matrix. Grain length/GL, grain width/GW, grain length to width ratio/GLWR, grain size/grain area/GS, grain circumferences/GC. The areas of ellipses showed the absolute value of corresponding correlation coefficients (*r*) (upper triangular). Right and left oblique ellipses and colors indicated positive and negative correlations, respectively. The values were corresponding *r* between the traits (lower triangular).

**Figure 3 ijms-23-01065-f003:**
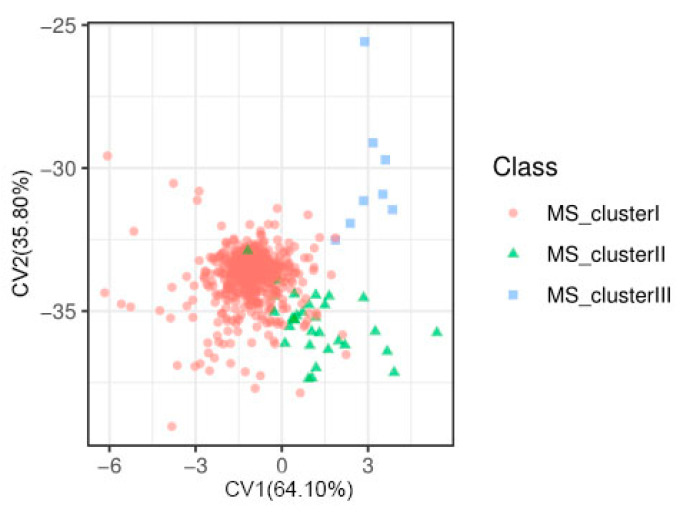
Graphic dispersion of scores in relation to axes representing the canonical variables for five traits related to grain shape (GL, GW, GLWR, GC, GS). Cluster I, cluster II, and cluster III are groups based on Mahalanobis distance between samples. Different colors represent different groups.

**Figure 4 ijms-23-01065-f004:**
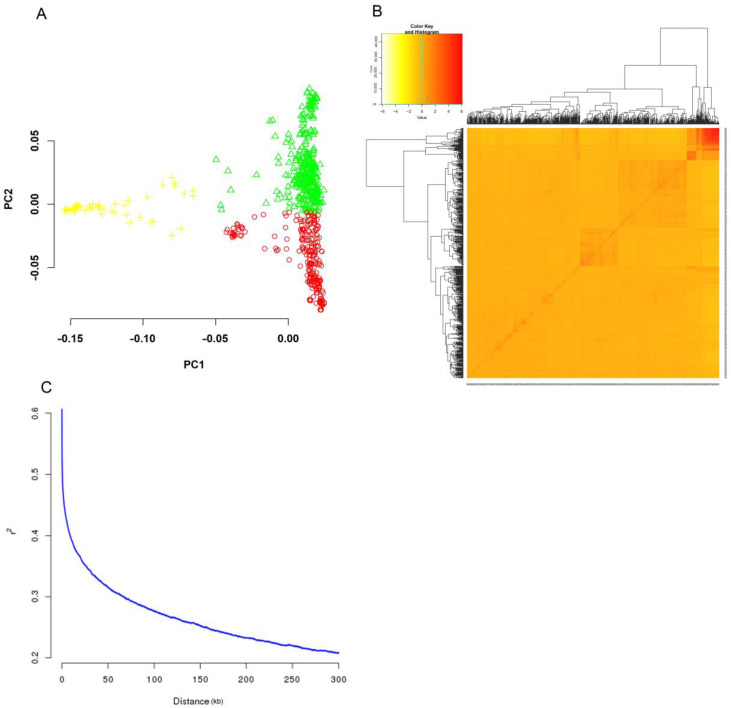
Population Structure, Kinship, and LD decay plot. (**A**) PCA plot for the 623 varieties. PC1 and PC2 indicate the score of principal components 1 and 2, respectively. Different colors represent different principal component score groups. (**B**) Heat map of kinship from R Package “pheatmap” shows the tree on the top and left. (**C**) LD decay. Y-axis was the average *r*^2^ value of each 5 kb region, and X-axis was the physical distance between markers.

**Figure 5 ijms-23-01065-f005:**
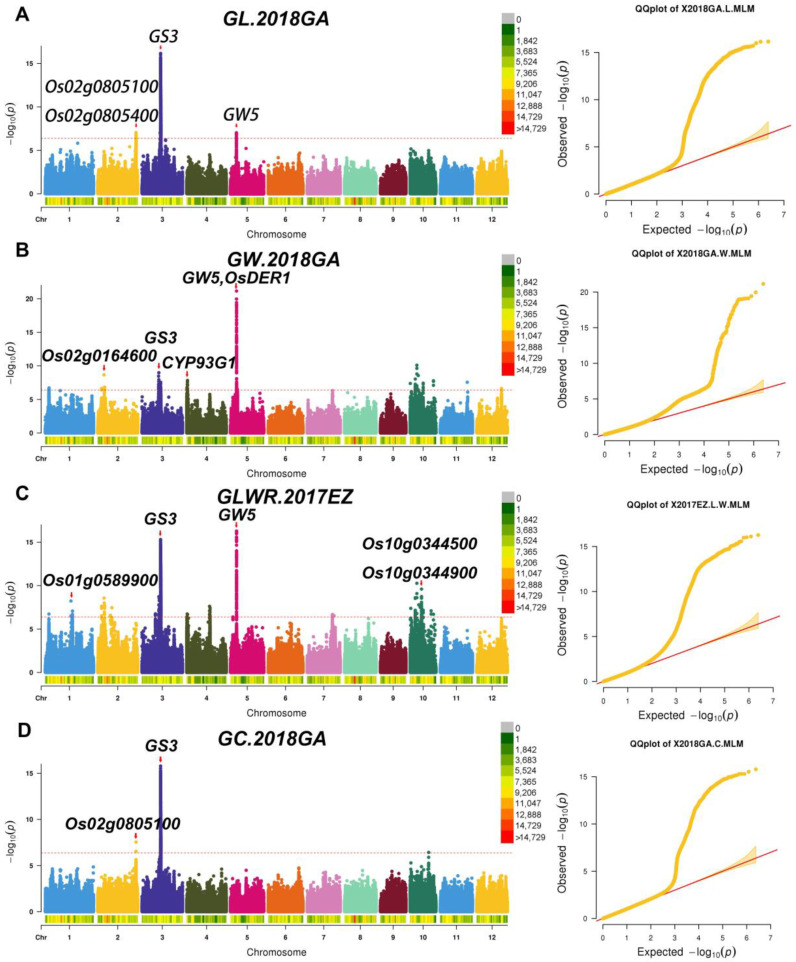
Manhattan plot and Q-Q plot. (**A**) 2018 GA’s GL Manhattan and Q-Q images. (**B**) 2018 GA’s GW Manhattan and Q-Q images. (**C**) 2017 EZ’s GLWR Manhattan and Q-Q images. (**D**) 2018 GA’s GC Manhattan and Q-Q images. The red arrow indicates the location of the gene. In the upper right corner is the SNP density indicator band, and different colors represent different SNP distribution densities. Gongan/GA, Ezhou/EZ.

**Figure 6 ijms-23-01065-f006:**
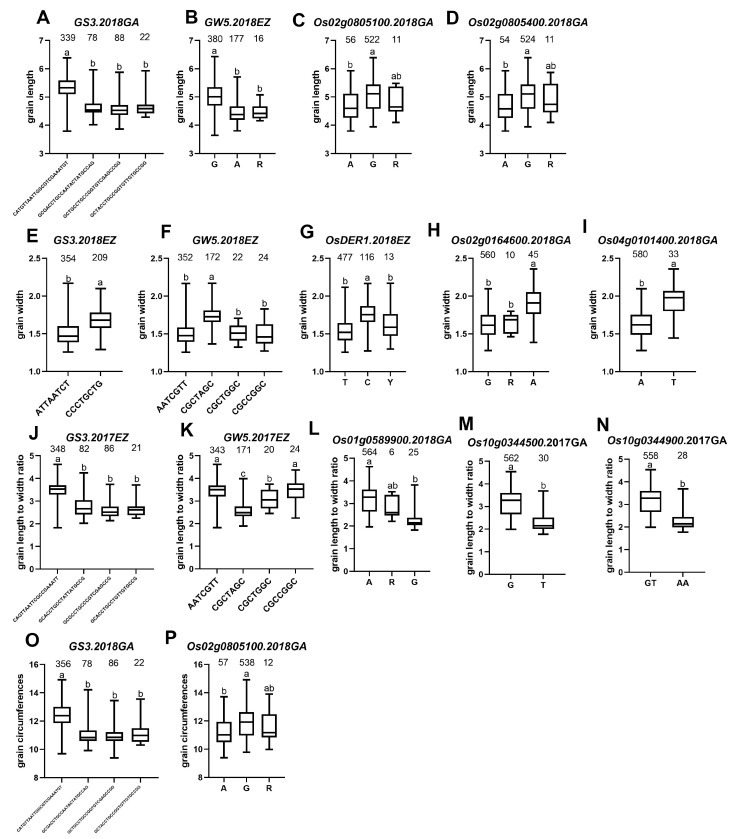
Haplotype box plot. (**A**–**D**) For grain length, haplotype box plot of (**A**) *GS3*, (**B**) *GW5*, (**C**) *Os02g0805100* and (**D**) *Os02g0805400*. (**E**–**I**) For grain width, haplotype box plot of (**E**) *GS3*, (**F**) *GW5*, (**G**) *OsDER1*, (**H**) *Os02g0164600,* and (**I**) *Os04g0101400*. (**J**–**N**) For grain length to width ratio, haplotype box plot of (**J**) *GS3*, (**K**) *GW5*, (**L**) *Os01g0589900*, (**M**) *Os10g0344500* and (**N**) *Os10g0344900*. (**O**,**P**) For grain circumferences, haplotype box plots of (**O**) *GS3* and (**P**) *Os02g0805100*. The letter (a, b, and c) suggested significance of ANOVA (for ≥three haplotypes) or t-test (for two haplotypes) at *p* < 0.01. The value on the box was the number of individuals of each haplotype. X-axis coordinates are the corresponding significant SNPs. Gongan/GA, Ezhou/EZ.

**Figure 7 ijms-23-01065-f007:**
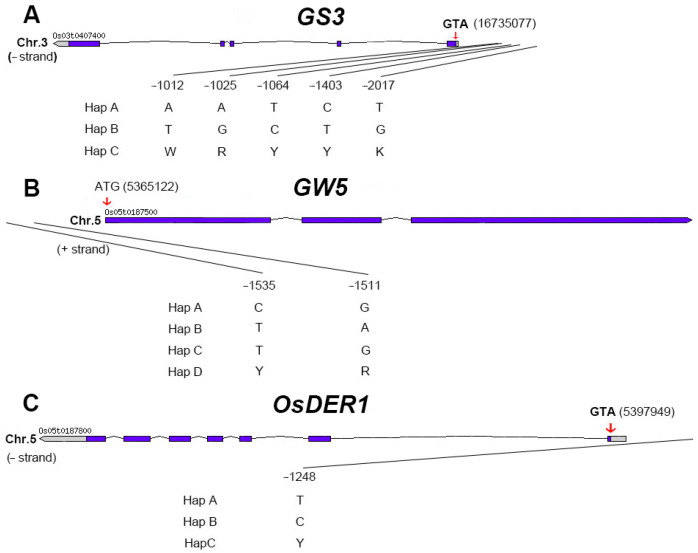
Gene structure, Significant SNP locus, and alleles. (**A**) *GS3*, Hap A (AATCT), Hap B (TGCTG) and Hap C (WRYYK). (**B**) *GW5*, Hap A (CG), Hap B (TA), Hap C (TG) and Hap D (YR). (**C**) *OsDER1*, Hap A (T), Hap B (C) and Hap C (Y). The red arrow indicates the location of the start codon “ATG”. The number after “ATG” is its position on the chromosome. The number on the significant SNP is its position relative to “ATG”.

**Figure 8 ijms-23-01065-f008:**
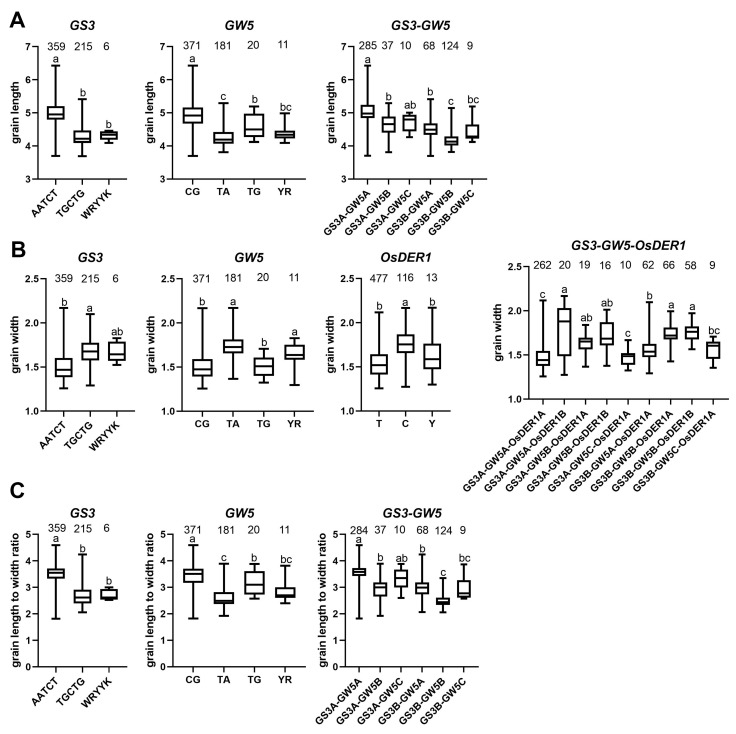
Box plot of alleles and allelic combinations. (**A**) grain length/GL, (**B**) grain width/GW, (**C**) grain length to width ratio/GLWR. The letter (a, b, and c) suggested significance of ANOVA at *p* < 0.01. The value on the box was the number of individuals of each allele. X-axis coordinates are the corresponding significant SNPs.

**Table 1 ijms-23-01065-t001:** Mahalanobis distances for phenotypic traits.

	GL	GW	GLWR	GC	GS
GL	−				
GW	−111,507.80	−			
GLWR	6039.21	−19,306.47	−		
GC	49,771.01	−588,205.40	78,805.02	−	
GS	7704.96	754,904.50	81,783.15	40,385.27	−

**Table 2 ijms-23-01065-t002:** Phenotypic heritability.

Env\Trait	GL	GW	GLWR	GC	GS
2017EZ	0.561198	0.587232	0.816784	0.435194	0.102882
2017GA	0.790264	0.842533	0.863557	0.750632	0.403186
2018EZ	0.417942	0.593973	0.780337	0.439132	0.220128
2018GA	0.682506	0.755747	0.802726	0.651486	0.519685
mean	0.6129775	0.69487125	0.815851	0.569111	0.31147025

**Table 3 ijms-23-01065-t003:** 39 important QTLs for GL, GW, GLWR, and GC detected in 2017GA, 2017EZ, 2018GA, and 2018EZ.

QTL	Env	Trait	CHRO	Position	Peak-SNP	Ref/Alt	Effect	SE	*P*	PVE(%)	Cloned Gene
*qGC2*	2018GA	GC	2	34214439-34472981	chr02_34347981	G/A	−0.555	0.099	2.88316 × 10^−8^	2.96	*OsmiR396a*; *OsmiR396c*
*qGC3*	2017GA	GC	3	16538239-17145970	chr03_16707611	T/C	−0.392	0.047	4.618 × 10^16^	42.07	*GS3*
	2017EZ	GC	3	16538239-17145970	chr03_16707611	T/C	−0.429	0.055	1.98333 × 10^−14^	41.50	*GS3*
	2018GA	GC	3	16538239-17145970	chr03_16692834	A/G	−0.488	0.057	1.64375 × 10^−16^	38.46	*GS3*
	2018EZ	GC	3	16538239-17145970	chr03_16669223	T/C	−0.518	0.060	4.16047 × 10^−17^	36.18	*GS3*
*qGC5*	2017GA	GC	5	5233756-5486894	chr05_5361894	G/A	−0.215	0.039	4.55499 × 10^−8^	33.45	*GW5; OsDER1*
*qGL2*	2018GA	GL	2	34202268-34492940	chr02_34339439	G/A	−0.239	0.044	9.10759 × 10^−8^	4.34	*OsmiR396a*; *OsmiR396c*
*qGL3.2*	2018GA	GL	3	16121544-16371544	chr03_16246544	G/A	0.110	0.020	9.6657 × 10^−8^	20.37	
*qGL3.3*	2017EZ	GL	3	16538239-17145970	chr03_16692706	C/T	−0.215	0.028	3.9741 × 10^−14^	41.52	*GS3*
	2018GA	GL	3	16538239-17145970	chr03_16692834	A/G	−0.236	0.027	7.18544 × 10^−14^	40.03	*GS3*
	2017GA	GL	3	16538239-17145970	chr03_16727804	G/A	−0.187	0.023	1.00333 × 10^−15^	39.81	*GS3*
	2018EZ	GL	3	16538239-17145970	chr03_16669223	T/C	−0.242	0.029	3.25938 × 10^−16^	35.72	*GS3*
*qGL5*	2017GA	GL	5	5233629-5496042	chr05_5361894	G/A	−0.132	0.020	3.72538 × 10^−11^	35.53	*GW5*; *OsDER1*
	2017EZ	GL	5	5233629-5496042	chr05_5361894	G/A	−0.145	0.024	1.49047 × 10^−9^	34.90	*GW5*; *OsDER1*
	2018EZ	GL	5	5233629-5488611	chr05_5359598	G/A	−0.138	0.026	9.59975 × 10^−8^	24.92	*GW5*; *OsDER1*
	2018GA	GL	5	5234598-5484598	chr05_5359598	G/A	−0.121	0.024	4.07439 × 10^−7^	27.56	*GW5*; *OsDER1*
*qGLWR1.1*	2017GA	GLWR	1	3182916-3432916	chr01_3307916	A/G	0.222	0.039	1.93411 × 10^−8^	0.57	
	2018EZ	GLWR	1	3182916-3432916	chr01_3307916	A/G	0.214	0.041	2.85427 × 10^−7^	0.38	
	2017EZ	GLWR	1	3182916-3432916	chr01_3307916	A/G	0.213	0.040	1.86665 × 10^−7^	0.32	
*qGLWR1.2*	2018EZ	GLWR	1	22885450-23135450	chr01_23010450	A/G	−0.287	0.052	4.5957 × 10^−8^	9.95	
	2018GA	GLWR	1	22885450-23135450	chr01_23010450	A/G	−0.276	0.052	1.61289 × 10^−7^	9.36	
	2017EZ	GLWR	1	22885450-23137896	chr01_23010450	A/G	−0.301	0.051	5.90519 × 10^−9^	9.03	
*qGLWR2.1*	2018EZ	GLWR	2	3328503-3578511	chr02_3453511	C/A	−0.180	0.035	2.76336 × 10^−7^	7.68	
	2017EZ	GLWR	2	3328503-3578581	chr02_3453511	C/A	−0.192	0.034	2.33924 × 10^−8^	7.39	
	2018GA	GLWR	2	3328511-3578511	chr02_3453511	C/A	−0.179	0.035	3.7586 × 10^−7^	7.16	
*qGLWR2.2*	2017GA	GLWR	2	5535710-5785710	chr02_5660710	T/A	−0.330	0.056	5.91233 × 10^−9^	6.60	
	2018GA	GLWR	2	5535710-5999873	chr02_5660710	T/A	−0.324	0.060	8.1989 × 10^−8^	6.34	
	2018EZ	GLWR	2	5535710-5999873	chr02_5874873	C/T	−0.373	0.071	2.23701 × 10^−7^	6.08	
	2017EZ	GLWR	2	5535710-6323430	chr02_5660710	T/A	−0.347	0.057	2.70846 × 10^−9^	5.92	
	2018EZ	GLWR	2	6073121-6323430	chr02_6198419	G/T	−0.338	0.064	1.90726 × 10^−7^	6.22	
	2018GA	GLWR	2	6073121-6323430	chr02_6198419	G/T	−0.358	0.065	4.58012 × 10^−8^	5.76	
*qGLWR2.4*	2017GA	GLWR	2	12281126-12531126	chr02_12406126	C/T	−0.313	0.058	1.20528 × 10^−7^	11.47	
*qGLWR2.5*	2018EZ	GLWR	2	12581170-12854381	chr02_12706170	G/A	−0.375	0.067	3.83638 × 10^−8^	12.35	
	2017GA	GLWR	2	12581170-12854381	chr02_12706170	G/A	−0.356	0.065	5.90344 × 10^−8^	12.21	
	2018GA	GLWR	2	12581170-12831170	chr02_12706170	G/A	−0.367	0.068	1.02856 × 10^−7^	11.60	
	2017EZ	GLWR	2	12581170-12854381	chr02_12729381	C/T	−0.377	0.067	3.44126 × 10^−8^	10.28	
*qGLWR2.6*	2018EZ	GLWR	2	13775664-14025664	chr02_13900664	A/G	−0.341	0.066	3.74571 × 10^−7^	11.84	
*qGLWR2.7*	2017GA	GLWR	2	15065202-15315202	chr02_15190202	C/T	−0.253	0.049	2.83929 × 10^−7^	10.74	
*qGLWR3.1*	2017GA	GLWR	3	11862123-12195220	chr03_12040893	G/A	−0.238	0.042	2.0198 × 10^−8^	10.29	
*qGLWR3.2*	2017GA	GLWR	3	14887077-15259083	chr03_15097804	T/C	−0.379	0.066	1.73085 × 10^−8^	11.94	
	2018EZ	GLWR	3	14972804-15262319	chr03_15137319	G/C	−0.370	0.063	8.16489 × 10^−9^	10.84	
	2017EZ	GLWR	3	14972804-15775796	chr03_15137319	G/C	−0.377	0.062	1.86381 × 10^−9^	9.87	
	2018GA	GLWR	3	15007020-15499312	chr03_15137319	G/C	−0.377	0.064	6.37505 × 10^−9^	10.27	
	2018GA	GLWR	3	15525398-15798206	chr03_15650795	C/T	−0.261	0.048	8.10411 × 10^−8^	9.26	
	2018EZ	GLWR	3	15525795-15798182	chr03_15673182	T/A	−0.253	0.049	2.89611 × 10^−7^	9.09	
*qGLWR3.3*	2018EZ	GLWR	3	16384313-17145970	chr03_16665078	G/A	−0.253	0.030	5.54904 × 10^−16^	38.74	*GS3*
	2017EZ	GLWR	3	16538239-17145970	chr03_16667236	A/C	−0.247	0.030	4.99029 × 10^−16^	40.92	*GS3*
	2017GA	GLWR	3	16538239-17144509	chr03_16665078	G/A	−0.242	0.030	1.84033 × 10^−15^	38.89	*GS3*
	2018GA	GLWR	3	16538239-17145970	chr03_16665078	G/A	−0.249	0.031	2.87413 × 10^−15^	38.73	*GS3*
*qGLWR4.4*	2018EZ	GLWR	4	20288335-20541326	chr04_20416326	G/A	−0.229	0.041	3.2422 × 10^−8^	7.48	
	2017EZ	GLWR	4	20288335-21006646	chr04_20416326	G/A	−0.226	0.040	2.42506 × 10^−8^	6.86	
	2018GA	GLWR	4	20291326-20541326	chr04_20416326	G/A	−0.220	0.041	1.22254 × 10^−7^	6.94	
	2018EZ	GLWR	4	20593177-21006646	chr04_20718177	G/A	−0.215	0.041	2.54712 × 10^−7^	5.45	
	2017GA	GLWR	4	20638861-21006958	chr04_20881646	G/A	−0.254	0.042	3.01979 × 10^−9^	5.45	
*qGLWR5.2*	2017EZ	GLWR	5	5231448-5561924	chr05_5361894	G/A	−0.214	0.025	5.32542 × 10^−17^	42.53	*GW5*; *OsDER1*
	2018GA	GLWR	5	5231448-5561924	chr05_5361894	G/A	−0.231	0.025	1.41282 × 10^−18^	41.88	*GW5*; *OsDER1*
	2017GA	GLWR	5	5231448-5503981	chr05_5359598	G/A	−0.210	0.024	5.38154 × 10^−17^	41.49	*GW5*; *OsDER1*
	2018EZ	GLWR	5	5231448-5561924	chr05_5359598	G/A	−0.232	0.025	1.19804 × 10^−18^	40.73	*GW5*; *OsDER1*
*qGLWR7.2*	2017GA	GLWR	7	23874546-24124546	chr07_23999546	G/T	−0.273	0.051	1.32149 × 10^−7^	10.23	
*qGLWR10.1*	2018EZ	GLWR	10	461202-711202	chr10_586202	A/T	−0.241	0.045	1.11028 × 10^−7^	12.01	
	2018GA	GLWR	10	461202-711202	chr10_586202	A/T	−0.237	0.045	1.8842 × 10^−7^	11.28	
	2017EZ	GLWR	10	461202-711202	chr10_586202	A/T	−0.253	0.045	4.17056 × 10^−8^	10.75	
*qGLWR10.2*	2017GA	GLWR	10	981239-1231239	chr10_1106239	G/T	−0.251	0.049	3.35516 × 10^−7^	10.74	
*qGLWR10.4*	2017GA	GLWR	10	3451996-3701996	chr10_3576996	G/A	−0.256	0.048	1.5324 × 10^−7^	10.03	
*qGLWR10.5*	2017EZ	GLWR	10	4480128-4738419	chr10_4612011	A/G	−0.276	0.046	2.50401 × 10^−9^	11.20	
	2017GA	GLWR	10	4487011-4737058	chr10_4612058	T/G	−0.283	0.046	1.62479 × 10^−9^	11.26	
	2018EZ	GLWR	10	4487011-4737058	chr10_4612058	T/G	−0.254	0.048	2.08188 × 10^−7^	10.79	
	2018GA	GLWR	10	4487011-4737058	chr10_4612058	T/G	−0.263	0.049	9.96257 × 10^−8^	10.62	
*qGLWR10.6*	2018EZ	GLWR	10	5900422-6150495	chr10_6025464	G/A	−0.304	0.053	1.57353 × 10^−8^	9.73	
	2017EZ	GLWR	10	5900422-6150524	chr10_6025464	G/A	−0.346	0.052	5.39359 × 10^−11^	9.46	
	2018GA	GLWR	10	5900422-6150495	chr10_6025464	G/A	−0.315	0.053	5.45461 × 10^−9^	9.25	
*qGLWR10.7*	2018EZ	GLWR	10	7593442-7843442	chr10_7718442	T/C	−0.313	0.058	8.71587 × 10^−8^	9.06	
	2018GA	GLWR	10	7593442-7843442	chr10_7718442	T/C	−0.311	0.058	1.33752 × 10^−7^	8.50	
	2017EZ	GLWR	10	7593442-7843442	chr10_7718442	T/C	−0.321	0.056	1.63274 × 10^−8^	8.11	
*qGLWR10.8*	2018EZ	GLWR	10	8856195-9106195	chr10_8981195	G/T	−0.332	0.062	1.05397 × 10^−7^	10.16	
	2018GA	GLWR	10	8856195-9106195	chr10_8981195	G/T	−0.332	0.062	1.42668 × 10^−7^	9.51	
	2017EZ	GLWR	10	8856195-9106195	chr10_8981195	G/T	−0.336	0.060	3.12832 × 10^−8^	9.06	
*qGLWR10.9*	2017EZ	GLWR	10	10025822-10458283	chr10_10223273	C/T	−0.381	0.059	2.51359 × 10^−10^	8.23	
	2017GA	GLWR	10	10042380-10458283	chr10_10272029	G/A	−0.363	0.061	4.73434 × 10^−9^	9.43	
	2018EZ	GLWR	10	10042380-10397029	chr10_10223273	C/T	−0.363	0.060	2.98474 × 10^−9^	9.16	
	2018GA	GLWR	10	10042380-10397029	chr10_10223273	C/T	−0.372	0.061	1.97669 × 10^−9^	8.79	
*qGW1.3*	2018EZ	GW	1	23933241-24512660	chr01_24304654	A/G	0.113	0.022	2.42535 × 10^−7^	6.23	*OsCTPS1*
*qGW2.1*	2017EZ	GW	2	3328503-3578511	chr02_3453511	C/A	0.069	0.012	1.13817 × 10^−8^	6.07	
	2018EZ	GW	2	3328503-3578511	chr02_3453511	C/A	0.065	0.012	2.15744 × 10^−7^	4.81	
	2018GA	GW	2	3328503-3578511	chr02_3453511	C/A	0.067	0.013	2.35387 × 10^−7^	4.12	
*qGW3.4*	2017GA	GW	3	16538324-17028634	chr03_16706516	T/C	0.053	0.008	5.96404 × 10^−11^	41.78	*GS3*
	2018EZ	GW	3	16540072-17040318	chr03_16706516	T/C	0.062	0.010	5.44229 × 10^−10^	30.86	*GS3*
	2017EZ	GW	3	16540424-16926472	chr03_16706516	T/C	0.056	0.009	3.95865 × 10^−9^	30.65	*GS3*
	2018GA	GW	3	16570517-16998965	chr03_16746142	A/G	0.059	0.011	2.73991 × 10^−8^	25.80	*GS3*
*qGW4.1*	2018GA	GW	4	11108-800893	chr04_628540	C/A	0.111	0.019	1.47095 × 10^−8^	8.52	*OsARG*
	2017EZ	GW	4	11113-261113	chr04_136113	G/A	0.091	0.017	1.77411 × 10^−7^	8.98	
	2018EZ	GW	4	263820-769400	chr04_644395	A/G	0.108	0.019	2.30805 × 10^−8^	7.48	*OsARG*
	2017EZ	GW	4	546523-796523	chr04_671523	C/T	0.099	0.019	3.60601 × 10^−7^	9.04	*OsARG*
*qGW5.1*	2017GA	GW	5	5231448-5561924	chr05_5359246	G/A	0.065	0.007	2.13646 × 10^−20^	42.12	*GW5*; *OsDER1*
	2018GA	GW	5	5231448-5585712	chr05_5359246	G/A	0.091	0.009	7.09101 × 10^−22^	36.43	*GW5*; *OsDER1*
	2018EZ	GW	5	5231448-5574689	chr05_5359681	C/T	0.082	0.009	5.765 × 10^−19^	36.22	*GW5*; *OsDER1*
	2017EZ	GW	5	5231448-5581997	chr05_5359246	G/A	0.071	0.008	7.27903 × 10^−17^	31.13	*GW5*; *OsDER1*
*qGW5.2*	2018GA	GW	5	5917480-6167480	chr05_6042480	C/T	0.043	0.008	3.08835 × 10^−7^	12.62	*JMJ703*
*qGW10.1*	2018GA	GW	10	52079-302079	chr10_177079	T/A	0.124	0.022	4.17434 × 10^−8^	8.97	*OsSCP46*
	2018EZ	GW	10	52079-302079	chr10_177079	T/A	0.120	0.022	5.35845 × 10^−8^	8.72	*OsSCP46*
*qGW10.7*	2017EZ	GW	10	5900422-6150495	chr10_6025464	G/A	0.126	0.019	1.49782 × 10^−10^	10.32	
	2018GA	GW	10	5900422-6150524	chr10_6025464	G/A	0.142	0.021	7.97234 × 10^−11^	8.94	
	2018EZ	GW	10	5900464-6150464	chr10_6025464	G/A	0.112	0.022	3.22894 × 10^−7^	7.82	
*qGW10.8*	2017EZ	GW	10	6986882-7439539	chr10_7297872	A/C	0.121	0.020	4.16284 × 10^−9^	10.05	
	2018GA	GW	10	7154870-7422872	chr10_7279870	C/G	0.108	0.018	2.7638 × 10^−9^	11.54	
*qGW10.12*	2017EZ	GW	10	10078190-10459267	chr10_10223273	C/T	0.122	0.021	9.81178 × 10^−9^	8.84	
	2018EZ	GW	10	10078190-10405532	chr10_10203190	A/G	0.127	0.022	1.04852 × 10^−8^	7.51	
	2017GA	GW	10	10147029-10418094	chr10_10272029	G/A	0.095	0.018	1.02675 × 10^−7^	7.89	

## Data Availability

The raw data for this study can be found in the BioProject ID PRJNA321462 (2016-05-13), PRJNA331215 (2016-07-25), and PRJNA734973 (2021-05-22) on NCBI and 3k RGP. The URL is https://www.ncbi.nlm.nih.gov/sra and https://registry.opendata.aws/3kricegenome/.

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
