# Peer review of "Identification and Allele Combination Analysis of Rice Grain Shape-Related Genes by Genome-Wide Association Study"

_ijms, 2022, doi:10.3390/ijms23031065_

Round 1

Reviewer 1 Report

Manuscript submitted by Meng et al. on “Identification and allele combination analysis of rice grain shape-related genes by genome-wide association study” performed a rigorous study to genetic control of grain shape in rice and identified seven new candidates including three cloned genes. This study can be useful in pyramiding genes for targeted grain shape. However, the following comments can help in improving the quality of the manuscript.

Major comments:

  1. Were the normality of the phenotypic data checked using statistical method (results line 100)? Include in the data analysis section in methods, how it was done?
  2. There are incomplete sentences especially in the materials and methods section. Use complete sentences.
  3. Include statistical analysis section and explain all the statistics used in the study
  4. English improvement must be done in the entire manuscript

Minor comments:

Line 53: Cite reference for “Many factors are affecting …”. Confirm if it is affecting or effect?

Line 57: Use regular numbers for numbering rather than number in symbol.

Line 63 – 85: This is redundant paragraph and can be removed. Only specific information related to the study should be retained to make it more concise and topic related.

Line 91: “excellent allele combination” what does this mean?

Line 93: “What's more” replace with more formal words

Line 129 -134: This includes background of the method and is not necessary in the manuscript and can be removed.

Figure 3. Include color index for figure 3A.

Line 322: “… with previous reports” mention the references

Line 336: Sentence not clear “Due to the interaction between …”

Line 340: delete “was”

Line 355: What does this mean “Repeat 3 times”

Line 412: What do you mean by “excellent allele”

Reviewer 2 Report

Paper "Identification and allele combination analysis of rice grain shape-related genes by genome-wide association study" is interesting.

Authors analysed a natural population containing 623 indica rice cultivars with rich genetic diversity for GWAS analysis. 
They combined rich phenotypic and genotypic data and adopted a strict MLM model for GWAS analysis. 

The results of GWAS were analyzed by various analysis methods. Why? Lack of comparison of these methods!!!

"GWAS analysis was performed using the R package "rMVP"." Methods should be described in more detail. "R package" is not informative.

Authors identified three cloned genes and seven new candidate genes.

Authors should use the canonical variate analysis and estimate Mahalanobis distances for phenotypic traits.

Paper needs major revision.

Round 2

Reviewer 1 Report

I would like to thank Meng et al. for making the suggested changes in the manuscript. Please pay attention to minor errors and providing high quality figures and tables. I do not have any further comments.

Author Response

Thank you again for your valuable comments on our manuscript. Your suggestions have greatly improved the quality of our articles.

Reviewer 2 Report

Text: "The operation steps of "rMVP" are as follows: (1) Data Preparation. Prepare phenotype and genotype data files and calculate population structure (Q) and kinship (K) based on the genotype files. (2) Data Input. Import the above four files into the operation. (3) Start GWAS. Important option parameters are model ("rMVP" offers three models: GLM, MLM, and FarmCPU.) and threshold (0.05 or 0.1, We chose 0.1). (4) Output. Here we have SNPs that are significantly correlated." is description of procedure but not methods!!! In the paper still lack of description of GWAS analysis.

Figure 3: This is not canonical variate analysis. Authors should be contact with statistician.

Paper need major reviosion.

Round 3

Reviewer 2 Report

Figure 3 not present results of canonical variate analysis. Authors should be contact with statistician.

Paper need major revision.

Round 4

Reviewer 2 Report

Now, all is ok.